# Roles of TrkC Signaling in the Regulation of Tumorigenicity and Metastasis of Cancer

**DOI:** 10.3390/cancers12010147

**Published:** 2020-01-08

**Authors:** Wook Jin

**Affiliations:** Laboratory of Molecular Disease and Cell Regulation, Department of Biochemistry, School of Medicine, Gachon University, Incheon 21999, Korea; jinwo@gachon.ac.kr

**Keywords:** TrkC, TrkC fusion, TrkC inhibitor, somatic mutation, targeted therapies

## Abstract

Tropomyosin receptor kinase (Trk) C contributes to the clinicopathology of a variety of human cancers, and new chimeric oncoproteins containing the tyrosine kinase domain of TrkC occur after fusion to the partner genes. Overexpression of TrkC and TrkC fusion proteins was observed in patients with a variety of cancers, including mesenchymal, hematopoietic, and those of epithelial cell lineage. Both microRNAs (miRNAs) and long non-coding RNAs (lncRNAs) were involved in the regulation of TrkC expression through transcriptional and posttranscriptional alteration. Aberrant activation of TrkC and TrkC fusion proteins markedly induces the epithelial-mesenchymal transition (EMT) program, growth rate, tumorigenic capacity via constitutive activation of Ras-MAP kinase (MAPK), PI3K-AKT, and the JAK2-STAT3 pathway. The clinical trial of TrkC or TrkC fusion-positive cancers with newly developed Trk inhibitors demonstrated that Trk inhibitors were highly effective in inducing tumor regression in patients who do not harbor mutations in the kinase domain. Recently, there has been a progressive accumulation of mutations in TrkC or the TrkC fusion protein detected in the clinic and its related cancer cell lines caused by high-throughput DNA sequencing. Despite given the high overall response rate against Trk or Trk fusion proteins-positive solid tumors, acquired drug resistance was observed in patients with various cancers caused by mutations in the Trk kinase domain. To overcome acquired resistance caused by kinase domain mutation, next-generation Trk inhibitors have been developed, and these inhibitors are currently under investigation in clinical trials.

## 1. Introduction

Trk proteins (Trk or NTRKs) are identified as members of tropomyosins fused to a tyrosine kinase domain and are single-pass transmembrane receptors. Trk proteins are activated by their neurotrophins, nerve growth factor (NGF), brain-derived neurotrophic factor (BDNF), neurotrophin-3 (NT-3), and neurotrophin 4/5 (NT4/5). TrkA, TrkB, and TrkC, encoded by *NTRK1*, *NTRK2*, and *NTRK3,* respectively, and neurotrophins, exhibit specificity in interactions with the specific receptors. TrkA preferentially binds NGF, and TrkB binds BDNF and neurotrophin-4/5, and TrkC physiologically binds to neurotrophin-3 as high-affinity transmembrane receptors for neurotrophins [1,2]. Moreover, a small peptide between the second immunoglobin-like C2 type 2 region and the transmembrane domain of Trk proteins affects ligand-binding specificity [3,4,5]. Neurotrophins and their-specific receptors regulate survival, growth, differentiation, and apoptosis in the peripheral and central neuronal systems. Activation of the Ras/MEK/MAPK pathway, PI3K/AKT pathway, and phospholipase C-gamma (PLCγ) signaling by Trk activation is crucial for neuronal survival [2,6,7].

The reduction of TrkC expression has been observed in neurodegenerative diseases, including Alzheimer’s (AD), Parkinson’s (PD), and Huntington’s diseases (HD). The selective degeneration and dysfunction of cholinergic basal forebrain neurons of the nucleus basalis is a feature of AD that mainly correlates with severe cognitive impairment. TrkC (58%) is well expressed in numerous NB of Meynert neurons in control brains, but these expressions were significantly reduced by about two-fold during progression (29.6%) in AD brains [8], and TrkC expression reduced considerably in cholinergic NB neurons during the progress of AD [9,10,11]. Moreover, TrkC expression, as well as NT-3, is remarkably expressed in the adult substantia nigra pars compacta, but reduced expression of TrkC in the SN of PD patients induced abnormal accumulation of α-synuclein as the hallmark of PD [12]. Moreover, TrkC expression restores long-term striatal depression on corticostriatal synaptic plasticity in the 3-NP-treated animal model of HD. TrkC activates the neuronal survival pathways, including the Ras/MEK/MAPK and PI3K/AKT pathways. Hence, TrkC-mediated activation of the Ras/MEK/MAPK and PI3K/AKT pathways promotes cellular functions such as proliferation, growth, and survival in cancer [13], raising the possibility that the role of TrkC protein provided from studies in the sympathetic nervous system may contribute to disease pathology.

## 2. Incidence of TrkC Expression in Cancer Development

In addition to the functional role of TrkC in the neuronal system, overexpression of TrkC is observed in many human tumors (Table 1). The involvement of TrkC in a variety of human cancers was first reported in studies on TrkC expression in neuroblastoma and glioma. Neuroblastoma is the most common extracranial solid tumor that occurs early childhood, and over 60% of the neuroblastomas are metastatic. It accounts for approximately 15% of pediatric cancer deaths [14]. In neuroblastoma, TrkC is highly expressed in 25% of primary neuroblastomas and is often accompanied by TrkA [15]. Moreover, a subset of stage IV neuroblastomas exhibits high-level NT-3 and TrkC co-expression [16]. In glioma, TrkC was up-regulated in 91.8% of glioma patient samples [17], and high-grade gliomas showed a more positive immunoreactivity than low-grade gliomas in NT-3 and TrkC expression [18]. Furthermore, TrkC was up-regulated in 86% of medulloblastomas and 68% of non-cerebellar primitive neuroectodermal (PNET) tumors (17 glial tumors, three ependymal tumors, and one teratoid tumor) [19].

TrkC is identified in several other types of human cancers as well. In the case of breast cancer, two studies have reported the incidence of TrkC in these tumor types. TrkC was more significantly overexpressed in basal-like breast cancer cells than in luminal cancer cells, and TrkC expression was elevated in 82% of breast cancer patients [20]. Hepatocellular carcinoma (HCC) represents approximately 90% of primary liver cancer and is the second main cause of cancer-related deaths in the world [21]. TrkC is overexpressed significantly in HCC cells, and its elevated expression was found to be correlated with the unmethylated TrkC promoter [22].

Moreover, TrkC expression was observed in 86% of tumors, in which TrkC molecules were present as alternatively spliced isoforms [23]. TrkC expression was also observed in primary and metastatic melanoma cells [24]. TrkC expression caused by immunoreactivity was observed in 62.5% of melanomas of various stages, and its expression significantly increased 58% of melanoma that progressed from in situ lesions, 91% of papillary dermal invasions, 57% of melanomas which invaded the deeper dermis, and 31% of melanomas which metastasized to sites other than the compound nevi [25]. Furthermore, there was an increase by 66% in the observed expression of TrkC in ductal pancreatic tissue compared to in normal adjacent tissue [26,27]. The occurrence of TrkC was observed at similar levels in prostate cancer specimens obtained from patients both with and without neoadjuvant hormonal therapy [28]. Moreover, TrkC expression was identified in lung cancer and leukemia. TrkC receptor was not detected in lung adenocarcinomas and bronchioloalveolar carcinomas, but was observed in 35% of well-differentiated squamous cell carcinomas within large size tumor cells, and in 55% of small cell lung cancers (SCLC) [29]. TrkC was significantly overexpressed in leukemia subtypes such as Lymphoblastic Leukemia, Acute Myeloid Leukemia (AML), and Chronic Lymphocytic Leukemia (CLL) [30]. Additionally, in gastric cancer (GC), high expression of TrkC was significantly correlated with distant metastasis, lymph node metastasis, distant metastasis, or recurrence of the disease [31]. 

Finally, during progression from typical thyroid C cell hyperplasia to the later stages of medullary thyroid carcinoma (MTC), substantial changes of TrkC were detected by using immunostaining. TrkC expression was not detected in healthy thyroid C cells, but moderate or strong TrkC immunostaining was identified in 87% of MTC tumors. Also, TrkC expression in MTC cell induces tumorigenic ability and primary tumor formation in nude mice [32].

### 2.1. The Functional Role of Long Noncoding RNA and microRNA in TrkC Expression 

Recent reports have demonstrated that miRNAs and lncRNAs are involved in the regulation of TrkC expression. miRNAs are 20–22 nucleotide long non-coding RNA molecules, which regulate gene expression at the post-transcriptional level. The expression of full-length TrkC is markedly reduced by miR-151-3p and miR-185. Also, expression of the truncated TrkC isoform was considerably reduced by miR-128, miR-485-3p, miR-765, and miR-768-5p [33]. Moreover, miR-9, miR-125a, and miR-125b were able to decrease cell growth, and truncated TrkC isoform was the target of miR-8, miR-125, and miR-125b in the repression of human neuroblastoma or medulloblastoma cell proliferation [34,35].

Additionally, lncRNAs are involved in the transcription of proteins directly and indirectly via transcriptional and posttranscriptional alteration, and are a potential therapeutic target of cancer [36]. Long noncoding RNA participates in the regulation of TrkC expression. Elevated expression of PVT1-214 is significantly associated with clinicopathological characteristics and poor survival of GC patients, and induces tumorigenic ability of GC by increasing TrkC expression via inhibition of miR-128 [37]. Another long noncoding RNA LINC00978 was markedly upregulated in GC patients and associated with poor survival outcome of GC patients. In addition, LINC00978 expression induces metastatic potential and inhibits apoptosis of cells. Moreover, LINC00978 promoted tumorigenicity and proliferation of cancer cells through inhibition of reduction of TrkC expression by miR-497 [38]. In contrast, TrkC was one of the target genes of LINC00052, and down-expression of TrkC increases the aggressiveness and proliferation of SMMC7721 cells. Also, LINC00052 suppresses the expression of the truncated isoform of TrkC by forming complementary base pairing with miR128, miR-485-3p, and miR-765 [39].

### 2.2. Somatic Mutations of TrkC in Cancer

Progressive accumulation of mutations can cause cancer or other diseases, and most tumors carry 1000 to 20,000 somatic point mutations, which has been reported in high-throughput DNA sequencing analyses of cancer cell genomes [40,41], with these mutations leading to constitutive activation of signaling circuits [42]. Thus, we now know about the mutation of TrkC identified in colorectal cancer (CRC) cell lines by mutational analysis of the tyrosine kinome, which was suggested to be a pathogenic mutation (G608S, I695V, R731Q, K732T, L760I) [43]. Also, TrkC has a nonsynonymous somatic mutation in pancreatic (H599Y, G608S, E322K) [44,45,46], breast (R678Q) [47], lung (H677Y, R721F) [48], and gastric (T149R, K746T, E543D) cancers [49]. Moreover, somatic mutations have been identified in 454 sites of TrkC, including the kinase domain, in 36 various cancer types (Table 2), and somatic mutation of the TrkC gene mainly occurs through missense mutations (Figure 1A). Around 25% of human cutaneous squamous cell carcinoma contains missense mutations in TrkC, and these missense mutations were also detected in ~7.8% of human lung adenocarcinomas and ~7% uterine endometrial carcinoma patients (Figure 1B). However, the origin of TrkC mutations remains unknown, but patients of various cancer types have a spectrum of many different TrkC mutations, including modifications of the tyrosine kinase domain. 

**Figure 1 cancers-12-00147-f001:**
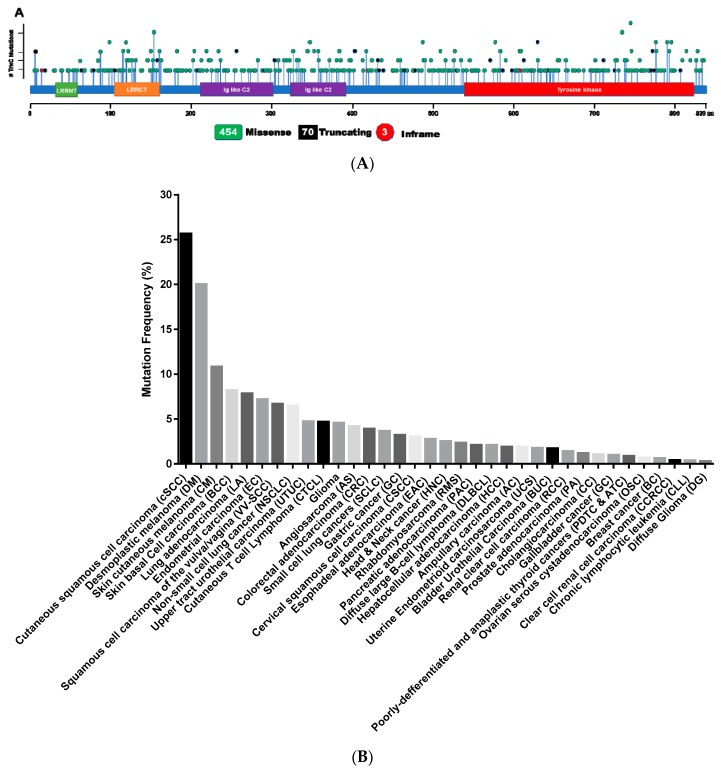
Somatic mutation frequency of TrkC in patients of 36 various cancer types. (**A**) Total of 530 mutations in TrkC, including 454 missense mutations, 70 truncating mutations, three inframe mutations, and discovered in patients of various cancers. Specifically, 177 of 530 mutations identified in the tyrosine kinase domain of TrkC. LRRNT: Leucine-rich repeat N-terminal domain, LRR1; Leucine-rich repeat region 1, LRR2: Leucine-rich repeat region 1, LRRCT: Leucine-rich repeat C-terminal domain, Ig-like C2: Immunoglobulin-C2-set domain, TM: transmembrane domain. (**B**) The overall frequency of mutated TrkC in cancer types. cSCC [50], DM [51], CM: TCGA Dataset, BCC: [52], LA: [53], EC: TCGA Dataset, VV-SCC: [54], NSCLC: [55], UTUC [56], CTCL: [57], AS: TCGA Dataset, CRC: [58], GC: TCGA Dataset, CSCC: TCGA Dataset, EAC: [59], HNC: TCGA Dataset, RMS: [60], PAC: [61], DLBCL: TCGA Dataset, HCC: TCGA Dataset, AC: Ampullary carcinoma [62], UCS: TCGA Dataset, BUC: TCGA Dataset, RCC: TCGA Dataset, PA: [63], CC: [64], GC: [65], PDTC & ATC: [66,67], OSC: TCGA Dataset, BC: [68], CCRCC: TCGA Dataset, CLL: [69], DG: [70].

**Table 1 cancers-12-00147-t001:** Detected TrkC in multiple histologies.

Cancer Name	Frequency	Ref.
Neuroblastoma	14/55 (25%)	[15]
Glioma	215/234 (91.8%)	[17]
Medulloblastomas	17/26 (85%)	[19]
Non-cerebellar primitive neuroectodermal (PNET) tumors	21/31 (67.7%)	[19]
Breast cancer	14/17 (82.4%)	[20]
Invasive ductal carcinoma	118/236 (49.6%)	[71]
Hepatocellular carcinoma	44/51 (86%)	[21]
Melanoma	40/64 (62.5%)	[25]
Pancreatic cancer	31/47 (66%)	[26,27]
Squamous cell carcinoma	3.5/10 (35%)	[29]
Small cell lung cancers	4.4/8 (55%)	[29]
Thyroid cancer	21.8/25 (87%)	[32]

**Table 2 cancers-12-00147-t002:** Mutation of TrkC in patients of Cancer types.

Cancer Name	Mutation (%)	Protein Change	Ref.
Ampullary Carcinoma (AC)	1.88	S136G, E211A, P796L	[62]
Angiosarcoma (AS)	4.17	P577L, G235E	TCGA
Oligodendroglioma (ODG)	0.68	Q808 *	[72]
Bladder Urothelial Carcinoma (BUG)	2.94	D609N, S477 *, L53F, E398K, G727 *, Q119E, E360K	TCGA
Breast Carcinoma (BC)	0.75	H349R, E810Q, G545A, L574V, E412K, N714K, P210T, A555V, K424T, G547R, V413M, Q177H, L108R, H254Y	[68]
Cervical Squamous Cell Carcinoma (CSCC)	3.04	A122T, F162S, L384M, D476N, L5V, R153Q, F162L	TCGA
Chronic Lymphocytic Leukemia (CLL)	0.33	A647T	[69]
Colorectal Adenocarcinoma (CRC)	4.17	F450L, L653I, S113A, R745Q, L115R, T777M, G608S, T149M, I759M, V217I, S117N, K181N, C782R, Q206H, T777M, T149M, K746R, K125N, R791Q, R745Q, Y456H, E86D, N191D, D624Y, Q159H, R201H, T730S, V97M, R745Q, G487D, R535M, R130H, T149M, A631D, D527G, Q586 *, V97M, R791W, G497R, Q145H, T332M, R130H, R791W, R814Q, A664T, K746T, R89H, Q119H, K346R	[58]
Cutaneous Melanoma (CM)	20	E762K, P509H, E211 *, H370N, P577S, L157F, E778K, D167N, R793 *, E318K, R542Q, E810K, H658Q, D697N, G235E, G356E, G633V, D584N, S359F, G356R, E590K, D576N, Y834N, R735C, G757R, R153L, G623E, R153L, W771 *, L299I, D242N, V217F, V37A, H423N, V726L, R116W, K768E, G178E, G727E, R735C, L152F, G339R, E819K, M667I, E86 *, Q773 *, W243L, Q255 *, P417L, D836H, T831I, P401S	TCGA
Cutaneous Squamous Cell Carcinoma (cSCC)	25.64	P577S, H264Y, D576N, Q773 *, E778K, G767E, A548V, D584N, L152F, D61G, G828E, K797E	[50]
Cutaneous T-Cell Lymphoma (CTCL)	4.65	P577S, S741N	[57]
Desmoplastic Melanoma (DM)	20	H128Y, P329L, E351K, S701F	[51]
Diffuse Glioma (DG)	0.38	I488T, L282R, I488V	[70]
Diffuse Large B-Cell Lymphoma (DLBCL)	2.08	D98N	TCGA
Endocervical Adenocarcinoma (EA)	1.94	E546K	TCGA
Esophageal Adenocarcinoma (EAC)	3.97	V687G, G487S, R46W, L115P, A826V	[59]
Gallbladder Cancer (GC)	3.13	I817M	[65]
Head and Neck Carcinoma (HNC)	2.51	H632Y, Q159K, S77 *, T253N, Q531R, A636V, H423N, Y705N, K367N, R326L, C362S, C231F, H729N,	TCGA
Hepatocellular Carcinoma (HCC)	2.47	G233S, Q145R, D527G, S701T, H622N, F395Y, E318D	TCGA
Cholangiocarcinoma (CC)	2.78	V451I	[64]
Lung Adenocarcinoma (LA)	7.83	L282M, P7R, H658N, H370N, L384M, N454S, S39R, Y188H, C320F, G545C, A380D, R121I, R814L, N718Y, P796S, G605L, G666C, G757V, N137K, P417H, R306H, M292I, T420S, H677Q, L639I, L827M, Y376C, P120H, S775 *, G487R, V97L, G67V, H84Y, G828V, K602N, P526Q, V779F, N382I, Y821F, K551N, N218H, E740K, R169L, P120H, T500S, S184R, D428H, S28Y, P330Q, T777K, K181N, A380D, F603I, R814L, E314D, S309I, P526Q, K397N, H394Q, K621N, G652V, G233V, I212T, P383A, R535M, N52K, F147L, V704F, G608C, V221L, W754C, E357D, Y604F, G463 *, E398Q, Y834F, R138L, Y821C, M202L, D240H, Q515H, K461R, V799L, Q773K, M464I, H729Y, E512K, P120H, V324A, Q172H, V273L, R343L, K346N, R121I, R459G, T506A, D495E, R343W, S741I, L364P, S4C, P509T, P612A, A435E, T230S, E314Q, H84N, G642 *, N338Y, T707K, D801N, V241A, G279A, G487S, F123L, S296R, L629F, G649C, A581P, Y744F, R343L, D635N, P738H, R735H, T563N	[53,55]
Pancreatic Adenocarcinoma (PA)	1.83	R153Q, V29M, R306H, K746T, E322K, Q643K, E223Q	[61]
Papillary Renal Cell Carcinoma (PRCC)	1.41	L270V, E179G, T490K, G104 *	TCGA
Plasma Cell Myeloma (PCM)	0.98	R745W, E351D	[73]
Prostate Adenocarcinoma (PA)	1.13	D609N, G497R, V640A, F747S, R793Q, P417L, T93M, P509S, T777M, T332M	[63]
Clear Cell Renal Cell Carcinoma (CCRCC)	2.86	D609N, R735H	TCGA
Rhabdomyosarcoma (RMS)	2.3	Y709F	[60]
Ovarian Serous Cancer (OSC)	0.69	P304L, L827F, D584E	TCGA
Skin Cancer, Non-Melanoma (Basal Cell Carcinoma; BCC)	8.19	S741N, S751R, G608I, E475K, R745W, Q673H, E590D, M99I, P467L, S117R, Q255 *, L760F, K381E, E154D, G696E, M202I, Q655R, R735C, K346R	[52]
Squamous Cell Carcinoma of the Vulva/Vagina (VV-SCC)	6.67	G437 *	[54]
Gastric cancer (GC)	3.18	H486N, H521N, K181R, T490M, L115P, R326H, R201H, D277G, W335R, S117T, R787H, R791W, L197F, A435E, L17 *	TCGA
Thyroid Cancer (TC)	0.85	R630W, N294T	[66,67]
Upper Tract Urothelial Carcinoma (UTUC)	4.71	D499N, D527Y, R153Q, R326H	[56]
Uterine Endometrioid Carcinosarcoma (UCS)	7.18	T261S, F617L, C523Y, T390I, M202I, K346N, R459W, P832S, A681T, D537Y, Y456H, R222 *, E598 *, R793Q, K111N, A96T, A580V, R222Q, E357D, H482Y, L187P, P55S, K125N, E556K, R47Q, S117N, A664T, G114R, G699D, I511T, S65N, K397N, I508T, E86D, N816S, E412K, V97M, G233D, G374D, Y352C, Q159H, A387T, E58K, D836A, V217A, D75N, M700I, R518C, E322K, D703N, S151N, F772L, V221I, L712P, E287D	TCGA
Esophageal Carcinoma (EC)	2.43	D277G, W335R, S117T, K181R, R791W, H486N, H521N, L197F, R787H, T490M	[59]

*: nonsense mutation; TCGA: TCGA database.

### 2.3. TrkC Fusion in Cancer

In most NTRK3 fusion proteins, the 3′ region of NTRK3, which contains the tyrosine kinase domain fused with the 5′ region of the partner gene, is expressed in cancer. ETV6-NTRK3 is known as a chimeric oncoprotein, which occurs in various cancer types, including mesenchymal, hematopoietic, and epithelial cell lineages. ETV6-NTRK3 was initially cloned and identified in five (100%) out of five congenital fibrosarcomas (CFS) and a pediatric spindle cell malignancy [74,75] in the case report of AML [76,77]. ETV6-NTRK3 expression was also identified in five of six cellular mesoblastic nephroma (CMN) [75,77] and 10 of 11 CMN [78], an infantile spindle cell tumor of the kidney. However, the expression of ETV6-NTRK3 was not detected in classical CMN and Wilm’s tumor, as there is recurrent chromosomal translocation (t(12;15)(p13;q25)). This rearrangement generates a gene fusion encoding the Helix-Loop-Helix Domain (HLH) of the ETV6 (TEL, ETS family transcription factor) linked to the tyrosine kinase (PTK) domain of TrkC [79].

In breast cancer, ETV6-NTRK3 is expressed in 12 (92%) of 13 patients with human secretory breast carcinoma (SBC), a rare subtype of IDC, and the resulting ETV6-NTRK3 protein functions as a chimeric protein with potent transforming activity in fibroblasts [80]. Moreover, secretory breast carcinomas are triple-negative and express basal markers, while secretory breast carcinomas with ETV6-NTRK3 protein belong to the basal-like breast carcinomas [81,82]. Moreover, ETV6-NTRK3 was identified in CRC [83,84], glioma [85,86], spitz tumor [87], lung adenocarcinoma [88], infantile fibrosarcoma [88,89], gastrointestinal stromal tumor [88,90], thyroid carcinoma [91,92], uterine sarcoma [86], and sinonasal adenocarcinoma [93]. ETV6-NTRK3 expression was detected in 75% of lung adenocarcinomas, 70% of infantile fibrosarcomas, and 49% of gastrointestinal stromal tumors [88]. Moreover, ETV6-NTRK3 was identified in 26% of papillary thyroid carcinomas [91,92], and interestingly, the prevalence in the rearrangement of ETV6-NTRK3 was associated with exposure to radiation based on a case study of patients who suffered from the Chernobyl accident. The rearrangement of ETV-NTRK3 significantly increased to 14.5% in radiation-related papillary thyroid carcinomas (PTCs) from an occurrence of 2% in sporadic PTCs [91]. The results of this study suggest that ETV6-NTRK3 may represent another type of chromosomal rearrangement associated with the robust growth pattern of PTC in patients exposed to radiation [91].

Additional NTRK3 fusion proteins, occurring in small numbers, have been identified in various cancer types. AKAP13-NTRK3 was identified in a rare case of low-grade glioma [94], and BTBD1-NTRK3 was induced in high-grade astrocytoma [85]. Another TrkC fusion, EML4-NTRK3, was identified in rare cases of several types of cancers, including uterine and vaginal sarcomas [95], dermatofibrosarcoma [96], infantile fibrosarcoma and congenital mesoblastic nephroma [97], infantile fibrosarcomas [98], and glioblastoma [84]. Moreover, the expression of EML4-NTRK3 induces the tumorigenic ability of NIH3T3 fibroblast cells in vivo and in vitro [99]. Interestingly, the majority of the TrkC fused to the 3′ region of HOMER2 contains WASP homology region 1 (WH1) and a coiled-coil domain. Moreover, TrkC combined with STRN or STRN3 as paralog of STRN includes a coiled-coil domain. STRN, a calmodulin-binding protein member, is a partner of ALK protein, and its fusion protein, which was identified in thyroid and lung carcinoma [100,101], leads to constitutive activation of ALK tyrosine kinase via dimerization mediated by the coiled-coil domain of STRN [100]. Moreover, most of the TrkC fusion proteins contain a coiled-coil domain, including STRN, STRN3, TFG, TPM4, HOMER2, MYH9, MYO5, and EML4 (Figure 2). There is a possibility that ligand-independent constitutive activation of tyrosine kinase domain of the TrkC fusion protein is mediated by the coiled-coil domain of the partner gene product. TrkC fusion partners have been recently identified in a wide range of cancer types (Figure 2 and Table 3).

### 2.4. The Biological Function of TrkC in Cancer

Four principal mechanisms mediate aberrant activation of receptor tyrosine kinase (RTK) in human cancers: autocrine activation, chromosomal translocations, overexpression, or gain-of-function mutations [122] and activation of signals by overexpression of oncogenes including RAS, RAF, and MYC would result in correspondingly increased tumorigenicity of cancer cells [123]. For example, the epidermal growth factor receptor (EGFR) was overexpressed in many cancers, and it was leading to overexpression and constitutive activation of EFGR tyrosine kinase activity [124,125]. Moreover, ErbB2, an EFGR family receptor, is highly overexpressed in various cancer types, including breast cancer, and promotes tumor progression via ligand-independent constitutive activation, while the overexpression correlates with poor prognosis [126,127]. As per previous reports, the primary mechanism of action by overexpression of TrkC occurs through increased TrkC activity without NT-3. Overexpression of TrkC in human MTC cells exhibited an increased growth rate, tumorigenic capacity, and primary tumor formation in vivo relative to control cells before ligand addition and after NT-3 addition, consistent with the fact that TrkC is phosphorylated before ligand addition [32]. Moreover, overexpression of TrkC in MCF10A breast cancer cells or RIE-1 normal intestinal epithelial cells increases the metastatic ability without treatment with NT-3 [20,128].

Auto-activation of RTKs recruited and activated a wide range of downstream signaling proteins via interaction with SH2 or PTB (phosphotyrosine binding domain) domain-containing signaling proteins [122,129]. Recent studies have reported that the tyrosine kinase domain of TrkC and ETV6-NTRK3 interacts with the SH2 domain of the Src and PTB domain of IRS-1. Ligand-independent PTK activation of ETV6-NTRK3 and TrkC in breast and colon cancer leads to constitutive activation of the Ras-MAP kinase (MAPK) mitogenic pathway and the phosphatidylinositol 3-kinase (PI3K)-AKT pathway, as well as upregulation of cyclin D1 mediating cell survival [111,113], In addition, its activation by TrkC and ETV6-NTRK3 is mediated via activation of c-Src by using complex formation [130,131]. IRS-1 functions as an adapter protein, linking ETV6-NTRK3 and TrkC for constitutive activation of downstream signals. After ETV6-NTRK3 associates with IRS-1, ETV6-NTRK3 recruits GRB2 and p85 and activates Ras-MAPK and PI3K-AKT pathways. However, mutation of the PTB activation loop tyrosine of ETV6-NTRK3 defects IRS-1-mediated transformation-associated pathways, including Ras-MAPK and PI3K-AKT in NIH3T3 cells by completely blocking phosphorylation of IRS-1 via failure to associated with IRS-1. These results indicate that the C-terminus, including the PTK domain of TrkC, is essential for transforming activity in multiple cell lineages [131,132].

Metastases produced by cancer cells are formed by a complex succession of an invasion-metastasis cascade from invading local epithelial cells in the surrounding extracellular matrix (ECM) to reinitiate neoplastic growths at metastatic sites [42]. In this cascade, the EMT induces a significant loss in junctional E-cadherins, which are essential cell-to-cell adhesion proteins that prevent dissociation of epithelial cells. Through activation of the EMT, the epithelial cells acquire mesenchymal features, which include motility, invasiveness, and heightened resistance to apoptosis [133]. Moreover, constitutive RTK activation and loss of E-cadherin lead to stabilization of the mesenchymal status of cancer cells and maintain continuous EMT-inducing heterotypic signaling from the tumor microenvironment [134]. TrkC enhances the metastatic potential of cancer via the induction of EMT in breast and colon cancers. TrkC induces JAK2 and STAT3 expression, but kinase-dead mutant of TrkC does not induce expression of these proteins. These results have revealed the constitutive activation of the JAK2-STAT3 signaling pathway triggered by the activation of tyrosine kinase of TrkC. Moreover, TrkC induces JAK2 stabilization through the suppression of JAK2 ubiquitination via inhibition of the JAK2-SOCS3 complex formation, activation of the EMT program through Twist-1 expression and the EMT transcription factor (EMT-TFs), and activation of the IL6/JAK2/STAT3 pathway [20,128].

Additionally, analysis has found that ETV6-NTRK3 upregulates genes that are associated with cell motion, membrane invagination, and regulation of cell proliferation, and also downregulates genes involved in cell adhesion [135].

The overexpression and activation of STAT1 were observed in several human cancers [136] and overexpression of STAT1 acquired chemoresistance and radioresistance in breast and lung cancers by suppressing the cytotoxic response and inducting prosurvival genes [137,138,139]. Interestingly, ETV6-NTRK3 attenuated STAT1 acetylation through STAT1 phosphorylation by direct interaction and eventually increased the activity of NF-κB by promoting nuclear translocation of NF-κB, through inhibition of the STAT1-NF-κB complex formation [135].

Transforming growth factor (TGF-β) is a multifunctional cytokine that functions as a potent growth inhibitor of epithelial cells and as a promoter of tumor progression. The property of growth inhibition by TGF-β depended on the signaling through cyclin-dependent kinase (CDK) inhibitors [140]. Recent results have highlighted the importance of PTK activation of TrkC that inhibits the growth inhibitory activity of TGF-β. Both TrkC and ETV6-NTRK3 directly interacted with TGF-β type II receptor and inhibited phosphorylation of TGF-β type II receptor, and eventually blocked both TGF-β-induced Smad2/3 phosphorylation and TGF-β growth inhibitory activity. Moreover, inhibition of the tyrosine kinase activity of TrkC or ETV6-NTRK3 restored TGF-β tumor suppressor activity [141,142].

Bone morphogenetic proteins (BMPs) closely resemble a bifunctional TGF-β in cancer cell regulation. TrkC blocks the BMP tumor suppressor activity through the suppression of bone morphogenetic protein 2 (BMP-2)-induced Smad1 phosphorylation and transcriptional activation by directly interacting with the BMP type II receptor [143]. Additionally, loss of REST function (transcription factor) as a tumor suppressor was identified in colon adenocarcinoma, lung cancer, and breast cancer. Moreover, the loss of REST expression in breast cancer significantly correlated with recurrence and poor survival of breast cancer patients. Additionally, the TSPY2/REST complex induces transcriptional repression of TrkC and restores the tumor inhibitory activity of TGF-β [144,145].

### 2.5. Targeted Therapies for Trk or Trk Fusion Protein

Trks are specifically intriguing due to the resulting chimeric oncoproteins which occur in various cancer types, and these fusion proteins are driven by ligand-independent constitutive activation, eventually activating downstream signaling pathways involved in proliferation, tumorigenicity, and the EMT in human cancers (Figure 3). This result is the reason why drug companies are developing small molecular inhibitors of TRK to treat cancer, arthritis, or pain-induced disease. Several inhibitors of Trk have been developed for the treatment of Trk or Trk fusion protein-mediated adult and pediatric solid tumors and hematologic malignancies.

#### 2.5.1. Larotrectinib

Larotrectinib (Vitrakvi, LOXO-101, Loxo Oncology Inc., Stamford, CT, USA and Bayer, Leverkusen, Germany) was first approved by the Food and Drug Administration (FDA) in November 2018 for pediatric, adult solid tumors that have Trk and Trk fusion protein without a known acquired resistance mutation, have no satisfactory alternative treatments, or have not progressed following treatment [146]. Larotrectinib is a small molecule which is orally-administered, and the most selective Trk tyrosine kinase inhibitor with half-maximal inhibitory concentration (IC_50_) levels in a low nanomolar range between 5–11 nmol/L and no activity against other kinase and non-kinase targets. It functions by inhibiting the autophosphorylation of Trk via binding to the ATP-bind site of Trk [146,147].

**Figure 3 cancers-12-00147-f003:**
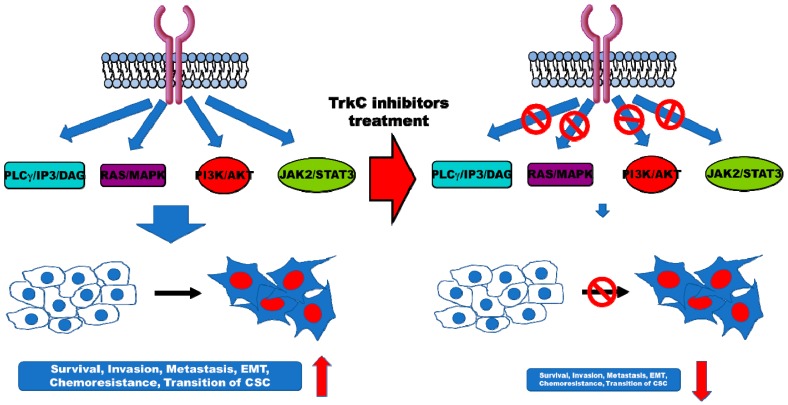
Diagram of TrkC biology and signaling identified in various cancers. TrkC and TrkC isoform can enhance and activate downstream of TRK signaling, including PLCγ/IP3/DAG [1,148,149], RAS/MAPK [1,79,132,148], PI3K/AKT [1,79,132,148], and JAK/STAT signaling [20,135], resulting in the promotion of survival, aggressiveness, chemoresistance, and generation of CSCs of cancer.

Larotrectinib is highly effective in inhibiting the proliferation of primary cancer cells, derived from patients harboring Trk fusion proteins. The IC_50_ was less than 100 mmol/L for CUTO-3.29 cells harboring MPRIP–NTRK1 and less than 10 nmol/L for KM12 cell harboring TPM3–NTRK1 or MO-91 cell harboring ETV6–NTRK3. Moreover, Larotrectinib inhibits tumor growth in a xenograft. Moreover, a phase I study of Larotrectinib for a woman with soft-tissue sarcoma metastatic to the lung demonstrated a marked improvement in multiple pulmonary metastases and almost complete tumor disappearance of the most extensive tumors [110]. Several clinical trials have assessed the efficacy and safety of Larotrectinib. The effectiveness of Larotrectinib was evaluated in 55 patients with Trk fusion-positive tumors, including TrkA (45% of the patients), TrkB (2%), and TrkC (53%), and with a recommended phase 2 dose of 100 mg twice daily in adults. The overall response rate (ORR) of patients according to the assessment was 75% (95% CI, 61–85). The complete response rate and the partial response rate was 22% and 53% of the ORR, respectively. At least 15% of the patients showed adverse events during treatment, which led to a dose reduction of Larotrectinib [88]. Moreover, Larotrectinib in a phase 1 trial significantly induced tumor regression in more than 90% of pediatric patients with solid tumors. Moreover, the efficacy of Larotrectinib for pediatric patients with TRK fusion cancers in phase 2 trials showed reductions in cancer occurrence. In total, 14 of 15 patients with Trk fusion-positive tumors in each of the Trk fusion proteins. TrkA (46% of the patients), *NTRK2* (6.7%), and *NTRK* (40%) showed objective responses at a median of 1.7 months. The maximum tolerated dose was estimated to be 100 mg/m^2^ of Larotrectinib [150]. Furthermore, the clinical trial of children with locally advanced TRK fusion sarcoma demonstrated that Larotrectinib induces a high response rate, including a reduction in the tumor [151]. In the case of a pediatric patient with ETV6-NTRK3 positive secretory breast cancer, treatment with Larotrectinib achieved an almost complete response and induced substantial tumor regression [152]. Additionally, the overall response rate (ORR) of ETV6-NTRK3 positive patients was 85% (95% CI, 64–96) [153].

#### 2.5.2. Entrectinib 

The FDA approved Entrectinib (Rozlytek, Gnentech Inc., South San Francisco, CA, USA) as a new Trk inhibitor for pediatric and adult solid tumors that have Trk, ROS proto-oncogene 1 (ROS1), and anaplastic lymphoma kinase (ALK) fusion proteins without a known acquired resistance mutation, and for tumors which are metastatic or for adults with metastatic NSCLC which are ROS1-positive. Entrectinib is a potent oral small-molecule inhibitor of Trk, ROS1, and ALK with IC_50_ values of 0.1 to 2 nM [154]. 

Entrectinib has been examined in several clinical trials. ALKA-372-001 phase I Trial showed that Entrectinib has a significant antitumor response in TrkA-positive CRC, ALK-rearranged neuroblastoma, and ROS1-, or ALK-positive NSCLC patients [155]. Moreover, the efficacy was assessed in patients with solid tumors with a Trk gene fusion in other phase trials (STARTRK-1 and STRATRK-2). 96% of the patients received 600 mg orally, once daily, had metastatic disease, and all the patients had Trk fusion proteins detected by Next-Generation Sequencing (NGS) and nucleic acid-based test. The ORR of patients according to the assessment was 57% (95% CI, 43–71), and the ORR by Trk fusion partners showed ENT6-NTRK3 (68%), TPM3-NTRK1 (50%), and TPP-NTRK1 (100%). Moreover, Entrectinib shows a partial response to LMNA-NTRK1 and SQSTM1-NTRK1 fusion proteins [156].

#### 2.5.3. Resistance to Larotrectinib and Entrectinib as Trk Inhibitor

Although there was a high overall response rate to Trk fusion proteins-positive solid tumors, acquired drug resistance was identified as a progressive disease after the administration of Trk inhibitors. The emergence of amino acid substitution in Trk represents a significant resistance mechanism against the Trk inhibitor. In the case of a patient with ETV6-NTRK3-positive mammary analog secretory carcinoma (MASC), treatment with Entrectinib showed a dramatic and durable response with an 89% reduction in tumor burden but revealed further disease progression in the right lower lobe of the lung. The NTRK3 G623R mutation mediated this case of resistance to Entrectinib. This alteration interfered with Entrectinib binding and conferred dramatically reduced sensitivity to Entrectinib inhibition, thereby increasing the IC_50_ value more than 250-fold. Moreover, NTRK3 G623R mutation conferred an increase in the IC_50_ value more than 500-fold for Larotrectinib and TSR-011 [157]. Additionally, LMNA-NTRK1-positive patients with metastatic colorectal cancer showed a remarkable response to Entrectinib, followed by resistance to Entrectinib mediated by NTRK1 G595R and G667C mutations and these mutations also showed immense resistance to Larotrectinib and TSR-011 [158].

Additionally, 11% of the patients showed further disease progression during the treatment with Larotrectinib after reported objective response or stable disease. Moreover, in tumor samples from 10 patients, amino acid substitution of the kinase domain was observed involving the solvent front position (TrkA G595R or TrkC G623R), the gatekeeper position (TrkA F589L), and the xDFG position (TrkA G667S or TrkC G696A) which eliminates unfavorable interactions by preventing inhibitor binding. As a result of this, eight out of 10 patients identified acquired resistance to Larotrectinib. Moreover, more than one mutation was identified in three patients [88]. These findings suggest that the biochemical characterization of more than 370 TrkC mutations, including a protein tyrosine kinase, is required to determine the potential mechanism of primary resistance to Trk inhibitors.

### 2.6. Next-Generation of Trk Inhibitor

Repotrectinib (TPX-0005) and LOXO-195 were developed to overcome acquired drug-resistance mutations, including solvent front and xDFG mutations of the kinase domain.

In terms of cell viability, Repotrectinib had a more potent IC50 value (<0.2 nmol/L) compared to that of Larotrectinib and Entrectinib. Moreover, TrkA G595R, TrkB G639R, TrkC G623R, and TrkC G623E exhibited a dramatic reduction in sensitivity to inhibition by Larotrectinib and Entrectinib, but Repotrectinib has a 42- and 62-fold increase, respectively, in the IC_50_ value compared to Larotrectinib and Entrectinib. Moreover, treatment with Repotrectinib for ETV6-NTRK3 and TrkC G623E mutation-positive patient with MASC showed a rapid and dramatic response to the drug and achieved regression within a few days of treatment [159].

LOXO-195 is another next-generation Trk kinase inhibitor designed to overcome recurrent resistance mediated by mutation of the gatekeeper position and the kinase domain (solvent front and xDFG). Treatment with LOXO-195 showed low inhibitory activity against TrkA G595R, TrkC G628R, and TrkA G667C, with IC_50_ values between 2.0 and 9.8 nmol/L. Moreover, LOXO-195 treatment of 2 patients who acquired resistance to Larotrectinib through TRKA G595R, and TRKC G623R, respectively, showed at least 90% inhibition of TRK target coverage at *C*_max_ and 50% inhibition at *C*_min_ of target pharmacokinetic thresholds (C_max_, C_min_). Furthermore, LMNA-NTRK1-positive colon cancer patients with TrkA G595R mutation displayed a rapid clinical response to therapy with LOXO-195. Moreover, ETV6-NTRK3 and TrkC G623R-positive pediatric patients with infantile fibrosarcoma showed visible tumor regression in the head and neck region [160]. Additionally, LOXO-195 is currently being evaluated in a Phase 1/2 trial. Of 29 evaluable patients via Phase I or FDA expanded access single patient protocol, 10 of these patients (34%) had a confirmed complete or partial response, and the ORR of patients with Trk kinase mutation according to the assessment was 45%. Moreover, nine out of 20 patients (45%) with a previous resistance to Larotrectinib treatment showed a complete or partial response to LOXO-195 [161].

## 3. Conclusions

TrkC and TrkC fusion proteins are associated with a variety of human cancers and play essential roles in the progression and metastasis of human cancers. The development of Larotrectinib and Rozlytek as Trk inhibitors has demonstrated a high overall response rate to Trk or Trk fusion proteins-positive solid tumors. However, acquired drug resistance was identified as a progressive disease by mutation, including mutations in the gatekeeper position and the kinase domain (solvent front and xDFG), still remains challenging, although Repotrectinib (TPX-0005) and LOXO-195 were developed to overcome kinase domain mutations and possesses potent and selective activity against TRKA G595R, TrkA G667C, and TRKC G623R. 

In patients of various cancer types, more than 530 different mutations of TrkC are identified, including a mutation in the tyrosine kinase domain (Figure 1A). These findings imply that mutations of TrkC in the kinase domain may lead to constitutive activation of TrkC, and this can lead to acquired resistance to first-generation or next-generation Trk inhibitors. To resolve this possibility, further studies will be required to determine the functional role of the TrkC mutation.

## Figures and Tables

**Figure 2 cancers-12-00147-f002:**
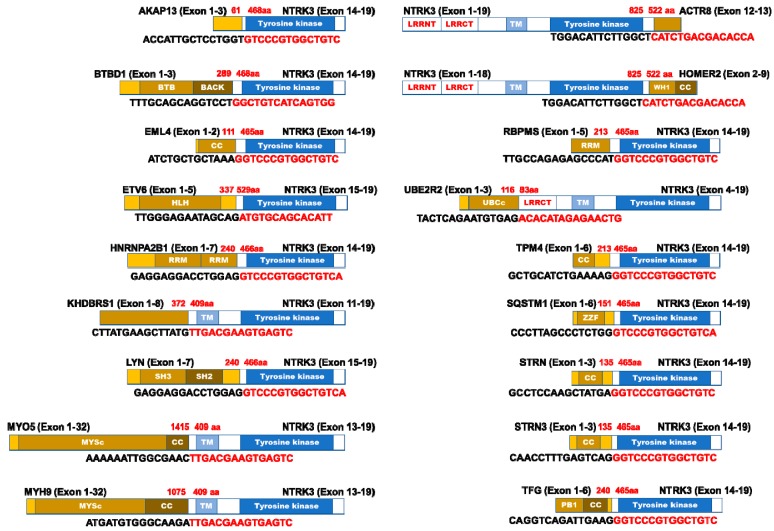
Diagram of TrkC fusion proteins identified in various cancers. In most TrkC fusion proteins, the carboxy-terminal of the TrkC protein, including the tyrosine kinase domain, fused with inframe to the amino-terminal binding partner. Nucleotide sequences indicate the breakpoint. CC: Coiled-coil, CC: Coiled-coil, HLH: Helix loop Helix, LRRCT: Leucine-rich repeat C-terminal domain, LRRNT: Leucine-rich repeat N-terminal domain, RRM: RNA recognition motif, BTB: protein-protein interaction, BACK: BTB and C-terminal Kelch, PB1: Phox and Bem1p, SH2: Src Homology 2, SH3: Src Homology 3, MYSc: Myosin motor, UBCs: Ubiquitin-conjugating enzyme E2, WH1: WASP homology region 1, ZnF: Zinc-binding, and TM: Transmembrane.

**Table 3 cancers-12-00147-t003:** Identification of TrkC fusion protein in cancer.

Fusion Protein	Chromosomal Location	Recurrent Chromosomal Translocation	Tumor Type
AFAP1-NTRK3	4p16.1	t(4;15)(p16;q25)	Glioblastoma [84]
AKAP13-NTRK3	15q25.3	t(15;15)(q25;q25)	Lung adenocarcinoma [102], Low-grade glioma [94]
BTBD1-NTRK3	15q25.2	t(15;15)(q25;q25)	Glioma [85]
CPEB1-NTRK3	15q25	t(15;15)(q25;q25)	Glioma [85]
EML4-NTRK3	2p21	t(2;15)(p21;q25)	Uterine and vaginal sarcomas [95], Dermatofibrosarcoma [96], Infantile fibrosarcoma and Congenital mesoblastic nephroma [97,99], Infantile fibrosarcomas [98], Glioblastoma [84].
ETV6-NTRK3	12p13.2	t(12;15)(p13;q25)	Congenital fibrosarcomas [74,75], Acute myeloid leukemia [76,77], Cellular mesoblastic nephroma [75,77,78], Secretory breast carcinoma [81,82], Colorectal cancer [83,84], Glioma [85,86], Spitz tumor [87], Lung adenocarcinoma [88], Infantile fibrosarcoma [88,89], Gastrointestinal stromal tumor [88,90], Thyroid carcinoma [103], Uterine sarcoma [86], Sinonasal adenocarcinoma [93], thyroid carcinomas [91,92], Mammary analog secretory carcinoma [104]
FAT1-NTRK3	4q35.2	t(4;15)(q35;q25)	Cervical squamous cell carcinoma (TCGA Dataset), [105]
HNRNPA2B1-NTRK3	7p15.2	t(7;15)(p15;q25)	Multiple myeloma [106]
KHDRBS1-NTRK3	1p35.2	t(1;15)(p35;q25)	Pediatric cutaneous congenital skin cancer [107]
LYN-NTRK3	8q12	t(8;15)(q12;q25)	Head and Neck squamous cell carcinoma (TCGA Dataset), [87,108]
MYH9-NTRK3	22q12.3	t(22;15)(q12;q25)	Spitz tumor [87]
MYO5A-NTRK3	15q21.2	t(15;15)(q21;q25)	Spitzoid tumor [87], Epithelioid melanocytic tumor [109]
NTRK3-HOMER2	15q25.2	t(15;15)(q25;q25)	Soft tissue sarcoma [110]
NTRK3-SCAPER	15q24.3	t(15;15)(q24;q25)	Epithelioid melanocytoma [111]
TPM4-NTRK3	19p13.12	t(19;15)(p13;q25)	Sarcoma [84]
ZNF710-NTRK3	15q26	t(15;15)(q26;q25)	Glioblastoma [84]
RBPMS-NTRK3	8p12	t(8;15)(p12;q25)	Glioma [112], Uterine Sarcoma [113], Thyroid carcinoma [66]
SPECC1L-NTRK3	22q11.23	t(22;15)(q11;q25)	Uterine sarcoma [86]
SQSTM1-NTRK3	5q35.3	t(5;15)(q35;q25)	Thyroid Cancer [114,115], Non-small-cell lung cancer [86,116]
STRN-NTRK3	2p22.2	t(2;15)(p22;q25)	Fibrosarcoma [117], Uterine sarcoma [118]
STRN3-NTRK3	14q12	t(14;15)(q12;q25)	Fibrosarcoma [117]
TFG-NTKR3	3q12.2	t(3;15)(q12;q25)	Solitary fibrous tumor [119,120]
UBE2R2-NTRK3	9p13.3	t(9;15)(p13;q25)	Multiple myeloma [106,121]
VIM-NTRK3	10q13	t(10;15)(q13;q25)	Thyroid carcinoma [86]
VPS18-NTRK3	15q15	t(15;15)(q15;q25)	Colon Adenocarcinoma (TCGA Dataset) [105]

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
