# Peer review of "Roles of TrkC Signaling in the Regulation of Tumorigenicity and Metastasis of Cancer"

_cancers, 2020, doi:10.3390/cancers12010147_

Round 1

Reviewer 1 Report

This manuscript presents an informative review regarding what has been uncovered through research with TrkC expression/signaling in the establishment and progression of a variety of cancers. The general organization of the manuscript shows merit, and there are no major issues with content.

However, there are major concerns with the overall writing and presentation of the material. If a significant re-edit of the manuscript occurs this could be of interest to may in the field, but in its current form it would be very challenging to decipher the information contained therein.

Below are a few examples of the issues mentioned above.

1) Needs significant editing for English grammar corrections. In its current state this manuscript is very difficult to read. The authors really need to proofread and edit carefully by reading through every sentence. There are many missing words, duplications and misused words in the text. 

For example: in the abstract

“ TrkC contributes to the clinicopathology of a variety of human cancers, and new chimeric oncoproteins containing tyrosine kinase domain of TrkC occur after fused to the partner genes.”

Should read: (need to give full name before abbreviating TrkC) TrkC contributes to the clinicopathology of a variety of human cancers, and new chimeric oncoproteins containing the tyrosine kinase domain of TrkC occur after being fused to the partner genes.

There are many instances where a left out word makes a sentence awkward and difficult to understand.

Another example: “ The pathological role of TrkC expression observed in neurological disorders and 40 neurodegenerative diseases, including Alzheimer’s (AD), Parkinson’s (PD), and Huntington’s (HD) 41 diseases.”

This is an incomplete sentence. What is it about the pathological role of TrkC are they trying to state? They do not finish the thought.

2) Make sure all abbreviations are fully named prior to using abbreviation

3) poor proof-reading example – duplication

Lines 50-52

These Ras/MEK/MAPK and PI3K/AKT pathways promote cellular functions such as proliferation, growth, survival in oncogenesis [13], raising the possibility that the role of TrkC protein provided from studies in the sympathetic nervous system may contribute to disease pathology.

Lines 53-55

These Ras/MEK/MAPK and PI3K/AKT pathways promote cellular functions such as proliferation, growth, survival in oncogenesis [13], raising the possibility that the role of TrkC protein provided from studies in the sympathetic nervous system may contribute to disease pathology.

4) Figure 1A need to be more clear – for instance – the authors claim that the figure shows 3 in-frame (red) and 3 other mutations (magenta) but even after magnifying the figure I could only find one circle that was not green or black…it looks magenta but is hard to tell. They should highlight the red and magenta more effectively to illustrate the point (maybe raise those above the black and green circles).

Fig 2 – font is too small to read at normal page size. Very difficult to examine nucleotide sequence and amino-terminal fusion gene names

5) line 290 – CDK – mislabeled as canonical dependent kinase – should be cyclin dependent kinase

6) the citations for Vitrakvi are confusing..there are facts stated that don’t seem to be supported with citations. The Rozlytek citations seem much more appropriately placed based on the text.

7) Format issue: Resistance to VITRAKVI and ROZLYTEK as Trk inhibitor: - why is this not a separately labeled section (2.5.3?) Does not flow well with the two inhibitors being described separately.

Author Response

Reviewer #1 Comments and Suggestions for Authors

This manuscript presents an informative review regarding what has been uncovered through research with TrkC expression/signaling in the establishment and progression of a variety of cancers. The general organization of the manuscript shows merit, and there are no major issues with content.

However, there are major concerns with the overall writing and presentation of the material. If a significant re-edit of the manuscript occurs this could be of interest to may in the field, but in its current form it would be very challenging to decipher the information contained therein.

Below are a few examples of the issues mentioned above.

1) Needs significant editing for English grammar corrections. In its current state this manuscript is very difficult to read. The authors really need to proofread and edit carefully by reading through every sentence. There are many missing words, duplications and misused words in the text. 

For example: in the abstract

“ TrkC contributes to the clinicopathology of a variety of human cancers, and new chimeric oncoproteins containing tyrosine kinase domain of TrkC occur after fused to the partner genes.”

Should read: (need to give full name before abbreviating TrkC) TrkC contributes to the clinicopathology of a variety of human cancers, and new chimeric oncoproteins containing thetyrosine kinase domain of TrkC occur after being fused to the partner genes.

There are many instances where a left out word makes a sentence awkward and difficult to understand.

Another example: “ The pathological role of TrkC expression observed in neurological disorders and 40 neurodegenerative diseases, including Alzheimer’s (AD), Parkinson’s (PD), and Huntington’s (HD) 41 diseases.”

This is an incomplete sentence. What is it about the pathological role of TrkC are they trying to state? They do not finish the thought.

We apologize for the grammatical errors and have edited our manuscript again. We have corrected and modified the sentence by the english language editing service, as suggested by the Reviewer.

2) Make sure all abbreviations are fully named prior to using abbreviation

We thank the Reviewer for this comment. As suggested by the Reviewer, all abbreviations fully named before using abbreviations.

3) poor proof-reading example – duplication                                                                        

Lines 50-52

These Ras/MEK/MAPK and PI3K/AKT pathways promote cellular functions such as proliferation, growth, survival in oncogenesis [13], raising the possibility that the role of TrkC protein provided from studies in the sympathetic nervous system may contribute to disease pathology.

Lines 53-55

These Ras/MEK/MAPK and PI3K/AKT pathways promote cellular functions such as proliferation, growth, survival in oncogenesis [13], raising the possibility that the role of TrkC protein provided from studies in the sympathetic nervous system may contribute to disease pathology.

 We apologize for this duplication, and we have corrected this, as suggested by the Reviewer.

4) Figure 1A need to be more clear – for instance – the authors claim that the figure shows 3 in-frame (red) and 3 other mutations (magenta) but even after magnifying the figure I could only find one circle that was not green or black…it looks magenta but is hard to tell. They should highlight the red and magenta more effectively to illustrate the point (maybe raise those above the black and green circles).

We thank the Reviewer for this comment. As suggested by the Reviewer, after magnified the figure 1A, we modified Figure 1A to make it clearer.

Fig 2 – font is too small to read at normal page size. Very difficult to examine nucleotide sequence and amino-terminal fusion gene names

 We thank the Reviewer for this comment and agree with the Referee’s opinion and suggestion. We magnified and modified figure 2 to make it more transparent. Also, we increased the font size.

5) line 290 – CDK – mislabeled as canonical dependent kinase – should be cyclin dependent kinase

We replaced it with the sentence suggested by the reviewer.

6) the citations for Vitrakvi are confusing..there are facts stated that don’t seem to be supported with citations. The Rozlytek citations seem much more appropriately placed based on the text.

We apologize for this confusion and have corrected this. We changed from Vitrakvi to Larotrectinbib for universal drug name, which usually used in the papers, and to match citations.

7) Format issue: Resistance to VITRAKVI and ROZLYTEK as Trk inhibitor: - why is this not a separately labeled section (2.5.3?) Does not flow well with the two inhibitors being described separately.

We apologize for this confusion and have corrected this. We separately labeled sections as 2.5.3, as suggested by the reviewer.

Reviewer 2 Report

In the review "Roles for TrkC signaling in the regulation of tumorigenicity and metastasis of cancer" the author nicely summarizes the current scope of knowledge of TrkC in carcinogenicity. While the work constitutes a comprehensive and extensive overview of the field, it is made hard to follow/read by the lack of basic grammar in every other sentence, with verbs and prepositions missing throughout. I have not marked every single instance of this throughout the text and it will require extensive re-editing for overall language.

Overall, some double checking of some facts is very much required with the wrong things referenced at times.

Finally, while this reviewer is a big fan of abbreviations there tend to be a few too many, being especially unnecessary when the term is never to be used again later and making some sentences overall unreadable.

Here are my specific comments by sections/lines:

Major comments:

It would probably be worthwhile introducing to the reader that TrkA, B, and C are encoded by NTRK1, 2, and 3, respectively, to make it easier for the reader to follow up on specific references used later in the manuscript

Paragraph of line 40-53: The link between abarrant TrkC expression and Ras/MEK/MAPK and Pi3K/Akt pathways is not explained, reference no 13 comes out of nowhere here and it is unclear on how the author came to this conclusion by reading the text. Please re-phrase/explain this further.

Section 2: " Incidence of TrkC expression in cancer development" The intro to this section is hard to read and would probably benefit from a table summarizing the incidence of TrkC expression in the respective cancer types, rather then the current text format, unless specific points about expression patterns of TrkC in specific cancer types really need to be made.

lines 106-109: This references states other numbers of mIRs that decrease neuroblastoma or medulloblastoma cell proliferation, are namely miR-9, mIR-125a and mIR-125b and NOT mIR-8 and mIR-125 as stated in the manuscript, see references 35 + 36.

line 120: its miR-128 NOT miR128-3p, see reference 40

Figure 1A: hard to read, could use some reformatting to not be so tiny

Figure 1B: would be easier to interpret if the cancers were spelled out in the graph X-axis labels and not just the figure legend.

Section 2.2, 2nd paragraph: the contents of this paragraph seem unnecessary, if the purpose is only to introduce the impact of constitutive activation of tyrosine kinase receptors in cancer development and linking it to TrkC being an important driver. This can be done in one sentence, without the extra information/references about unrelated TKRs which are not needed here.

Figure 2: is a very nice summary of NTRK3 fusion proteins, however it would be nice to include which cancers these occur in next to each schematic and not only in the seperate, hard to read table 2.

line 246-248: This sentence seems to be a bit at odds with the previous/disconjointed to the point that is trying to be made with the previous sentence. This needs to be re-worded if you want to get your point across, of ligand binding being unnecessary when looking at the effect of TrkC overexpression on tumorgenicity. 

section 2.5.1 Vitrakvi: Vitrakvi is a brand name and the inhibitor should probably be primarily be referred to as Lacrotrectinib... same for section 2.5.2 Rozlytek (Entrectinib)

lines 337-340: What was the response rate in patients carrying the NTRK3 fusion genes? i.e. TrkC fusion proteins around which this review is centered? This should probably be mentioned here.

Minor comments:

line 13: formatting error "noncoding"

lines 51-53 and lines 54-56 are the same sentence repeated.

line 94: formatting error "TrkC"

line 188-190: wording of this sentence should reflect the reference more accurately, stating that it was found that the fusion protein was expressed in 12 of 13 cases of this rare IDC subtype, not in all SBC ever analysed, also overall the sentence is hard to read. So is the following summarizing reference no 63, please re-phrase

line 206-207: formatting error

Reference no 73+74 are the same reference.

line 237-238: formatting error " Epidermal growth factor receptor"

line 245: formatting error "in vivo"

line 268: formatting error "The"

line 290: CDK stands for cyclin dependent kinases not canonical dependent kinases

line 299: maybe mention that REST is a transcription factor

line 305-309: this sentence needs rephrasing to be readable

line 348: formatting error: "rozlytek"

line 352: formatting error: "Efficacy"

line 354: formatting error: "All"

Author Response

Reviewer #2 Comments and Suggestions for Authors

In the review "Roles for TrkC signaling in the regulation of tumorigenicity and metastasis of cancer" the author nicely summarizes the current scope of knowledge of TrkC in carcinogenicity. While the work constitutes a comprehensive and extensive overview of the field, it is made hard to follow/read by the lack of basic grammar in every other sentence, with verbs and prepositions missing throughout. I have not marked every single instance of this throughout the text, and it will require extensive re-editing for overall language.

We apologize for the grammatical errors and have edited our manuscript again. We have corrected and modified the sentence by the english language editing service, as suggested by the Reviewer.

Overall, some double checking of some facts is very much required with the wrong things referenced at times.

We apologize for the wong reference and have edited our manuscript again, as suggested by the reviewer.

Finally, while this reviewer is a big fan of abbreviations there tend to be a few too many, being especially unnecessary when the term is never to be used again later and making some sentences overall unreadable.

As suggested by the reviewer, we deleted unnecessary abbreviations, which never to be used again later.

Here are my specific comments by sections/lines:

Major comments:

It would probably be worthwhile introducing to the reader that TrkA, B, and C are encoded by NTRK1, 2, and 3, respectively, to make it easier for the reader to follow up on specific references used later in the manuscript

As suggested by the reviewer, we inserted this sentence in the paragraph of line 31.

Paragraph of line 40-53: The link between abarrant TrkC expression and Ras/MEK/MAPK and Pi3K/Akt pathways is not explained, reference no 13 comes out of nowhere here and it is unclear on how the author came to this conclusion by reading the text. Please re-phrase/explain this further.

We apologized for the wong reference and corrected it as suggested by the reviewer. We have also inserted this sentence regarding the link between aberrant TrkC expression and Ras/MEK/MAPK and Pi3K/Akt pathways briefly.

Section 2: " Incidence of TrkC expression in cancer development" The intro to this section is hard to read and would probably benefit from a table summarizing the incidence of TrkC expression in the respective cancer types, rather then the current text format, unless specific points about expression patterns of TrkC in specific cancer types really need to be made.

We apologize for the inconvenience and have edited our manuscript again to understand well, as suggested by the reviewer. Also, We summarized incidence of TrkC in various cancers as Table 1.

lines 106-109: This references states other numbers of mIRs that decrease neuroblastoma or medulloblastoma cell proliferation, are namely miR-9, mIR-125a and mIR-125b and NOT mIR-8and mIR-125 as stated in the manuscript, see references 35 + 36.

We apologized for this error, and we have corrected it.

line 120: its miR-128 NOT miR128-3p, see reference 40

We apologized for this error, and we have corrected it.

Figure 1A: hard to read, could use some reformatting to not be so tiny

We thank the Reviewer for this comment. As suggested by the Reviewer, after magnified the figure 1A, we modified Figure 1A to make it clearer.

Figure 1B: would be easier to interpret if the cancers were spelled out in the graph X-axis labels and not just the figure legend.

We magnified and modified figure 1B to make it more transparent. Also, we labeled cancer names in the X-axis of the graph (Figure 1B), as suggested by the Reviewer. Moreover, we modified figure legend of figure 1B.

Section 2.2, 2nd paragraph: the contents of this paragraph seem unnecessary, if the purpose is only to introduce the impact of constitutive activation of tyrosine kinase receptors in cancer development and linking it to TrkC being an important driver. This can be done in one sentence, without the extra information/references about unrelated TKRs which are not needed here.

We thank the Reviewer for this comment. As suggested by the Reviewer, we delete unnecessary paragraph to remove the extra information and references which are unrelated to TrkC information.

Figure 2: is a very nice summary of NTRK3 fusion proteins, however it would be nice to include which cancers these occur in next to each schematic and not only in the seperate, hard to read table 2.

We thank the Reviewer for this comment. As suggested by the Reviewer, we try to combine the figure and cancer names, but it is hard to combine because it has no space to include which cancers these occur in next to each schematic. So, instead of combined figure2 and table 3, we modified table 3 to understand and accessible to reading.

line 246-248: This sentence seems to be a bit at odds with the previous/disconjointed to the point that is trying to be made with the previous sentence. This needs to be re-worded if you want to get your point across, of ligand binding being unnecessary when looking at the effect of TrkC overexpression on tumorgenicity. 

As suggested by the reviewer, we modified this sentence to explain more clearly.

section 2.5.1 Vitrakvi: Vitrakvi is a brand name and the inhibitor should probably be primarily be referred to as Lacrotrectinib... same for section 2.5.2 Rozlytek (Entrectinib)

We apologize for this confusion and have corrected this. We changed from Vitrakvi and Rozlytek to Larotrectinbib and Entrectinib for universal drug names, which usually used in the papers, and to match citations.

lines 337-340: What was the response rate in patients carrying the NTRK3 fusion genes? i.e. TrkC fusion proteins around which this review is centered? This should probably be mentioned here.

We inserted a paragraph with the response rate in patients carrying the ETV6-NTRK3 fusion gene in line 327.

Minor comments:

line 13: formatting error "noncoding"

We apologized for this error, and we have corrected it.

lines 51-53 and lines 54-56 are the same sentence repeated.

We apologized for this error, and we have corrected it.

line 94: formatting error "TrkC"

We apologized for this error, and we have corrected it.

line 188-190: wording of this sentence should reflect the reference more accurately, stating that it was found that the fusion protein was expressed in 12 of 13 cases of this rare IDC subtype, not in all SBC ever analysed, also overall the sentence is hard to read. So is the following summarizing reference no 63, please re-phrase

We apologize for the wong reference and have edited our manuscript again, as suggested by the reviewer. Also, we modified the sentence to understand easily.

line 206-207: formatting error

We have corrected it.

Reference no 73+74 are the same reference.

We have corrected it.

line 237-238: formatting error " Epidermal growth factor receptor"

We have corrected it.

line 245: formatting error "in vivo"

We have corrected it.

line 268: formatting error "The"

We have corrected it.

line 290: CDK stands for cyclin dependent kinases not canonical dependent kinases

We have corrected it.

line 299: maybe mention that REST is a transcription factor

As suggested by the reviewer, we inserted this sentence.

line 305-309: this sentence needs rephrasing to be readable

As suggested by the reviewer, we modified this sentence to explain more clearly.

line 348: formatting error: "rozlytek"

We have corrected it.

line 352: formatting error: "Efficacy"

We have corrected it.

line 354: formatting error: "All"

We have corrected it.

Reviewer 3 Report

A very complete review however i think is missing a final figures that combine with the title or different figures in each subtitles such as: 2.4 and 2.5.

Author Response

Reviewer #3 Comments and Suggestions for Authors

A very complete review however i think is missing a final figures that combine with the title or different figures in each subtitles such as: 2.4 and 2.5.

Thank you for your assistance.          

As suggested by the reviewer, we made figures (Figure 3) to explain sections 2.4 and 2.5.

Round 2

Reviewer 1 Report

The citation list needs to be formatted in the final form to match the font of the manuscript.

Author Response

Comments and Suggestions for Authors

The citation list needs to be formatted in the final form to match the font of the manuscript.

Thank you for your suggestion. We formatted in the final form to match the form of the manuscript.

Reviewer 2 Report

The extensive editing that followed the first review process is reflected in the current version of the manuscript and has greatly improved its readability and overall structure. There are, however, still some points that need to be addressed prior to publishing:

Major comments still relating to content:

(1) Reference no 136 is inappropriate for Figure 1. This document is not only not a scientific reference, it also contains no mention of NSCLC. or its somatic mutation frequency in patients. Reference no 136 is further used in a wrong on pages 10 and 11 of table 3 citing it as a reference for multiple myeloma gene fusions, which is not in this document.

correct reference here for multiple myeloma at lest is: Taylor J, Pavlick D, Yoshimi A, Marcelus C, Chung SS, Hechtman JF, et al. Oncogenic TRK fusions are amenable to inhibition in hematologic malignancies. J Clin Investig. 2018;128:3819–25. PMC6118587

(2) Reference no 139: This document is not only not a scientific reference, furthermore it is used in figure 1 and table 3 to reference the following points:  

figure 1: prostate cancer somatic mutation frequency, this information is not in this reference -> please supply the correct reference here

table 3: LYN-NTRK3 in head and neck squamous cell carcinoma, this information is not in this reference -> please supply the correct reference here

table 3: RBPMS-NTRK3 in thyroid cancer, this information is not in this document -> please supply appropriate reference

(3) Overall references in table 3: I have taken 3 references at random throughout table no 3 and they are all wrong or even missing information

Example 1: Spitz tumours gene fusions are referenced as no. 68, which is fact no. 87 in the revised document

Example 2: glioblastoma gene fusions are referenced as no. 79, which is fact no. 101 in the revised document, furthermore there are 3 gene fusions listed here, where in fact the reference lists 4, please supply a reasoning here why next to AFAP1-NTRK3, EML4-NTRK3 and ZNF710-NTRK3, the in the paper stated BCR-NTRK3 was not listed.

Example 3: reference no 148 for listing SPECC1L-NTRK3 for uterine sarcoma and VIM-NTRK3 for thyroid carcinoma, doesn't exist anymore in the revised document (or the one previously to be precise, which was previously missed).

Please revise your reference numbers in table 3 (and double check thought entire document, including figures), as they are likely following the 3 examples all wrong and some details missing, which at this stage is unacceptable.

The new figure 3 is a little confusing, as it implies that using TrkC inhibitors will lead to survival, invasion, metastasis, etc. Maybe rethink its organization and please include at least some references where the involvement of upregulation of PLCy/IP3/DAG, RAS/MAPK, Pi3K/Akt and Jak2/STAT3 pathways have been specifically shown to be upregulated following TrkC mutations or fusion events. This information is generally missing throughout the manuscript and will need some specific references/evidence, if it is to remain a strong point throughout, in the abstract and as its own figure.

Minor comments:

overall not all references contains doi links, this needs to be done consistently, either include it for all or for none. The following still lack a doi, or need some re-editing: reference no.: 4-6, 16, 22, 27, 30, 35 & 38(formatting), 77, 79, 80, 89 (formatting), 101, 110, 112, 115 (formatting and missing doi), 118 (formatting), 129 (formatting), 133 (formatting and missing doi), 134, 135, 138 (formatting), 144 please check the journals referencing style there are inconsistencies throughout in terms of how many authors are listed as a minimum for each reference, please review.

Author Response

Comments and Suggestions for Authors

The extensive editing that followed the first review process is reflected in the current version of the manuscript and has greatly improved its readability and overall structure. There are, however, still some points that need to be addressed prior to publishing:

Major comments still relating to content:

(1) Reference no 136 is inappropriate for Figure 1. This document is not only not a scientific reference, it also contains no mention of NSCLC. or its somatic mutation frequency in patients. Reference no 136 is further used in a wrong on pages 10 and 11 of table 3 citing it as a reference for multiple myeloma gene fusions, which is not in this document.

correct reference here for multiple myeloma at lest is: Taylor J, Pavlick D, Yoshimi A, Marcelus C, Chung SS, Hechtman JF, et al. Oncogenic TRK fusions are amenable to inhibition in hematologic malignancies. J Clin Investig. 2018;128:3819–25. PMC6118587

We apologized for our mistake. We found that the reference of Figures and Tables in the manuscript was not connected to EndNote software for managing citations and references. So, we updated to the correct references.

(2) Reference no 139: This document is not only not a scientific reference, furthermore it is used in figure 1 and table 3 to reference the following points:  

figure 1: prostate cancer somatic mutation frequency, this information is not in this reference -> please supply the correct reference here

table 3: LYN-NTRK3 in head and neck squamous cell carcinoma, this information is not in this reference -> please supply the correct reference here

table 3: RBPMS-NTRK3 in thyroid cancer, this information is not in this document -> please supply appropriate reference

We apologized for our mistake. We found that the reference of Figures and Tables in the manuscript was not connected to EndNote software for managing citations and references. So, we updated to the correct references.

(3) Overall references in table 3: I have taken 3 references at random throughout table no 3 and they are all wrong or even missing information

Example 1: Spitz tumours gene fusions are referenced as no. 68, which is fact no. 87 in the revised document

Example 2: glioblastoma gene fusions are referenced as no. 79, which is fact no. 101 in the revised document, furthermore there are 3 gene fusions listed here, where in fact the reference lists 4, please supply a reasoning here why next to AFAP1-NTRK3, EML4-NTRK3 and ZNF710-NTRK3, the in the paper stated BCR-NTRK3 was not listed.

Example 3: reference no 148 for listing SPECC1L-NTRK3 for uterine sarcoma and VIM-NTRK3 for thyroid carcinoma, doesn't exist anymore in the revised document (or the one previously to be precise, which was previously missed).

Please revise your reference numbers in table 3 (and double check thought entire document, including figures), as they are likely following the 3 examples all wrong and some details missing, which at this stage is unacceptable.

We apologized for our mistake. We found that the reference of Figures and Tables in the manuscript was not connected to EndNote software for managing citations and references. So, we updated to the correct references.

The new figure 3 is a little confusing, as it implies that using TrkC inhibitors will lead to survival, invasion, metastasis, etc. Maybe rethink its organization and please include at least some references where the involvement of upregulation of PLCy/IP3/DAG, RAS/MAPK, Pi3K/Akt and Jak2/STAT3 pathways have been specifically shown to be upregulated following TrkC mutations or fusion events. This information is generally missing throughout the manuscript and will need some specific references/evidence, if it is to remain a strong point throughout, in the abstract and as its own figure.

We apologize for this confusion and have corrected this. As your suggestion, we modified the figure 3. Also, we added the references in figure 3. Moreover, we showed that ETV6-NTRK3 upregulates RAS/MAPK and PI3K/AKT pathway in Line 236-248. Furthermore, we showed that ETV6-NTRK3 attenuated STAT1 acetylation by direct interaction in Line 269-135. Additionally, we showed upregulation of PLCg activation by ETV6-NTRK3 with reference no. 148.

Minor comments:

overall not all references contains doi links, this needs to be done consistently, either include it for all or for none. The following still lack a doi, or need some re-editing: reference no.: 4-6, 16, 22, 27, 30, 35 & 38(formatting), 77, 79, 80, 89 (formatting), 101, 110, 112, 115 (formatting and missing doi), 118 (formatting), 129 (formatting), 133 (formatting and missing doi), 134, 135, 138 (formatting), 144 please check the journals referencing style there are inconsistencies throughout in terms of how many authors are listed as a minimum for each reference, please review.

As your suggestion, we added doi in all references. Also, we formatted with the MDPI referencing style.  

Round 3

Reviewer 2 Report

The current revised version of the manuscript has greatly improved in overall readability and structure. Here are some final details that need addressing:

Major comments:

line 52-54: "Hence, TrkC-mediated activation of Ras/MEK/MAPK and PI3K/AKT pathways promote cellular functions such as proliferation, growth, survival in cancer [13,14]" Reference no 13 is not appropriate here, as there is zero mention of TrkC expression in cancer or activation of the aforementioned pathways:

[13] Muragaki, Y.; Timothy, N.; Leight, S.; Hempstead, B.L.; Chao, M.V.; Trojanowski, J.Q.; Lee, V.M. Expression of trk receptors in the developing and adult human central and peripheral nervous system. J Comp Neurol 1995, 356, 387-397, doi:10.1002/cne.903560306.

line 70-72: "TrkC was significantly overexpressed in basal-like breast cancer cells than in luminal cancer cells, and TrkC expression was elevated in 82% of breast cancer patients [21] and 49.6% of patients with invasive ductal carcinoma (IDC) [22]." Reference no. 22 actually states that high expression of TrkC is favourable for the outcome of IDC, contrary to the picture being painted in the review article that high TrkC equals a growth factor for cancer... this need to be mentioned by author if this reference is to stay here

line 81-86: "Furthermore, TrkC expression was mainly found in pancreatic cancer tissues (66%) than in healthy pancreatic tissue and observed in both benign and malignant prostate tissues. The intensity of TrkC expression correlated with the TNM stage. TrkC expression was more induced at advanced tumor stages (stage III and IV) [28,29]. The occurrence of TrkC was observed at similar levels in prostate cancer specimens obtained from patients both with and without neoadjuvant hormonal therapy [30]." Several things in this section, Ref 29, authors state that there was an increase in 66% in the observed expression of TrkC in ductal pancreatic tissue compared to normal adjacent tissue, please state it in this manner in the text. Also in Ref 29 the authors state that the ligand NT3 expression correlated with stage III&IV and such a case could not be made for the receptor TrkC, so this statement is wrong in the review text.

line 116-117: "Additionally, LINC00052 suppresses the expression of the truncated isoform of TrkC by forming complementary base pairing with miR128, miR-485-3p, and miR-765 [41]" In this article the authors find that downregulation of TrkC actually increases the invasion and proliferation of their hepatocarcinoma cell line. This needs to be stated in the review and not glossed over, otherwise this is a biased representation of the literature.

line 122-124: "Thus, we now know about the mutation of TrkC identified in colorectal cancer (CRC) cell lines by mutational analysis of the tyrosine kinome, which is suggested to be a pathogenic mutation (H599T, G608S, I695V, R731Q, K732T, L760I) [45,46]" Mutation H599T is not mentioned in ref 45 or supplement, i.e. not found officially there in colorectal cancer. Ref 46 states this, but also a cases in lung and breast, so either mention both or in none, please don't mix and match.

Minor comments:

line 76-77: "Moreover, TrkC expression was observed in 86% of tumors, in whichTrkC molecules were present as alternatively spliced isoforms [25]" Formatting and grammar error

line 279-282: "Trks are specifically intriguing due to the resulting chimeric oncoproteins which occur in various cancer types, and these fusion proteins are driven by ligand-independent constitutive activation, eventually activating downstream signaling pathways involved in proliferation, tumorigenicity and the EMT in human cancers (Figure 3)." typo

line 295: Figure 3: greatly improved, slight formatting error with the text sizes in the respective pictograms, maybe reduce size to fit.

line 793-796: formatting of reference 133:"Lannon, C.L.; Martin, M.J.; Tognon, C.E.; Jin, W.; Kim, S.J.; Sorensen, P.H.B. A highly conserved NTRK3 C-terminal sequence in the ETV6-NTRK3 oncoprotein binds the phosphotyrosine binding domain of insulin receptor substrate-1. An essential interaction for transformation. (vol 279, pg 6225, 2004). J Biol Chem 2004, 279, 15706-15706, doi: 10.1074/jbc.M307388200."

line 848-850: formatting of reference 153: "Laetsch, T.W.; DuBois, S.G.; Mascarenhas, L. Larotrectinib for paediatric solid tumours harbouring NTRK gene fusions: phase 1 results from a multicentre, open-label, phase 1/2 study (vol 19, pg 705, 2018). Lancet Oncol 2018, 19, E229-E229, doi: 10.1016/S1470-2045(18)30119-0.

Author Response

Comments and Suggestions for Authors

The current revised version of the manuscript has greatly improved in overall readability and structure. Here are some final details that need addressing:

Major comments:

line 52-54: "Hence, TrkC-mediated activation of Ras/MEK/MAPK and PI3K/AKT pathways promote cellular functions such as proliferation, growth, survival in cancer [13,14]" Reference no 13 is not appropriate here, as there is zero mention of TrkC expression in cancer or activation of the aforementioned pathways:

[13] Muragaki, Y.; Timothy, N.; Leight, S.; Hempstead, B.L.; Chao, M.V.; Trojanowski, J.Q.; Lee, V.M. Expression of trk receptors in the developing and adult human central and peripheral nervous system. J Comp Neurol 1995356, 387-397, doi:10.1002/cne.903560306.

We apologize for the mistake. As your suggestion, we deleted the sentence related to Reference No. 13.

line 70-72: "TrkC was significantly overexpressed in basal-like breast cancer cells than in luminal cancer cells, and TrkC expression was elevated in 82% of breast cancer patients [21] and 49.6% of patients with invasive ductal carcinoma (IDC) [22]." Reference no. 22 actually states that high expression of TrkC is favourable for the outcome of IDC, contrary to the picture being painted in the review article that high TrkC equals a growth factor for cancer... this need to be mentioned by author if this reference is to stay here

We thank the Reviewer for this comment. As your suggestion, when we checked again, this statement was contrary to the picture being painted in our review article. So, we deleted the sentence related to Reference No. 22.

line 81-86: "Furthermore, TrkC expression was mainly found in pancreatic cancer tissues (66%) than in healthy pancreatic tissue and observed in both benign and malignant prostate tissues. The intensity of TrkC expression correlated with the TNM stage. TrkC expression was more induced at advanced tumor stages (stage III and IV) [28,29]. The occurrence of TrkC was observed at similar levels in prostate cancer specimens obtained from patients both with and without neoadjuvant hormonal therapy [30]." Several things in this section, Ref 29, authors state that there was an increase in 66% in the observed expression of TrkC in ductal pancreatic tissue compared to normal adjacent tissue, please state it in this manner in the text. Also in Ref 29 the authors state that the ligand NT3 expression correlated with stage III&IV and such a case could not be made for the receptor TrkC, so this statement is wrong in the review text.

We thank the Reviewer for this comment and we apologize for the mistake. As your suggestion, we changed sentences such as there was an increase in 66% in the observed expression of TrkC in ductal pancreatic tissue compared to normal adjacent tissue. Also, like your mention, we can not found any correlation between TrkC expression and tumor stage. we deleted this sentence “The intensity of TrkC expression correlated with the TNM stage. TrkC expression was more induced at advanced tumor stages (stage III and IV)”.

line 116-117: "Additionally, LINC00052 suppresses the expression of the truncated isoform of TrkC by forming complementary base pairing with miR128, miR-485-3p, and miR-765 [41]" In this article the authors find that downregulation of TrkC actually increases the invasion and proliferation of their hepatocarcinoma cell line. This needs to be stated in the review and not glossed over, otherwise this is a biased representation of the literature.

As your suggestion, we added the sentence “In contrast, TrkC was one of the target genes of LINC00052, and down-expression of TrkC increase aggressiveness and proliferation of SMMC7721 cells.”.

line 122-124: "Thus, we now know about the mutation of TrkC identified in colorectal cancer (CRC) cell lines by mutational analysis of the tyrosine kinome, which is suggested to be a pathogenic mutation (H599T, G608S, I695V, R731Q, K732T, L760I) [45,46]" Mutation H599T is not mentioned in ref 45 or supplement, i.e. not found officially there in colorectal cancer. Ref 46 states this, but also a cases in lung and breast, so either mention both or in none, please don't mix and match.

We thank the Reviewer for this comment and we apologize for the mistake. As you mentioned, we found that mutation of H599Y is not mentioned in Ref 45 referred to Ref 46. H599Y mentioned at Ref 46 but we can not found Ref 45. So, as your suggestion, we deleted the sentence (H599Y) related to Reference No. 45. Also, we check whether the H599Y of TrkC was detected in breast and lung cancers in Ref 46 but we can not found the H599Y of TrkC in Refs (breast and lung cancers) of Ref 46. We only found that H599Y was detected in pancreatic cancer. So, we modified sentences.

Minor comments:

line 76-77: "Moreover, TrkC expression was observed in 86% of tumors, in whichTrkC molecules were present as alternatively spliced isoforms [25]" Formatting and grammar error

We apologize for the grammatical errors and have corrected our manuscript again.

line 279-282: "Trks are specifically intriguing due to the resulting chimeric oncoproteins which occur in various cancer types, and these fusion proteins are driven by ligand-independent constitutive activation, eventually activating downstream signaling pathways involved in proliferation, tumorigenicity and the EMT in human cancers (Figure 3)." Typo

We apologize for the grammatical errors and have corrected our manuscript again.

line 295: Figure 3: greatly improved, slight formatting error with the text sizes in the respective pictograms, maybe reduce size to fit.

As your suggestion, we modified the Figure 3 and have reduced size to fit again.

line 793-796: formatting of reference 133:"Lannon, C.L.; Martin, M.J.; Tognon, C.E.; Jin, W.; Kim, S.J.; Sorensen, P.H.B. A highly conserved NTRK3 C-terminal sequence in the ETV6-NTRK3 oncoprotein binds the phosphotyrosine binding domain of insulin receptor substrate-1. An essential interaction for transformation. (vol 279, pg 6225, 2004)J Biol Chem 2004, 279, 15706-15706, doi: 10.1074/jbc.M307388200."

We apologize for the error and have corrected our manuscript again.

line 848-850: formatting of reference 153: "Laetsch, T.W.; DuBois, S.G.; Mascarenhas, L. Larotrectinib for paediatric solid tumours harbouring NTRK gene fusions: phase 1 results from a multicentre, open-label, phase 1/2 study (vol 19, pg 705, 2018)Lancet Oncol 2018, 19, E229-E229, doi: 10.1016/S1470-2045(18)30119-0.

We apologize for the error and have corrected our manuscript again.